# An Enhanced Method for the Use of Reptile Skin Sheds as a High-Quality DNA Source for Genome Sequencing

**DOI:** 10.3390/genes14091678

**Published:** 2023-08-25

**Authors:** Yeyizhou Fu, Yan Zhuang, Shu-Jin Luo, Xiao Xu

**Affiliations:** 1The State Key Laboratory of Protein and Plant Gene Research, School of Life Sciences, Peking University, Beijing 100871, China; fyyz1901111501@pku.edu.cn (Y.F.); z75y20@163.com (Y.Z.); 2Peking-Tsinghua Center for Life Sciences, Academy for Advanced Interdisciplinary Studies, Peking University, Beijing 100871, China; 3Peking-Tsinghua-NIBS (PTN) Program, Academy for Advanced Interdisciplinary Studies, Peking University, Beijing 100871, China

**Keywords:** DNA extract method, reptiles, shed skins, whole genome resequencing

## Abstract

With the emergence of high-throughput sequencing technology, a number of non-avian reptile species have been sequenced at the genome scale, shedding light on various scientific inquiries related to reptile ecology and evolution. However, the routine requirement of tissue or blood samples for genome sequencing often poses challenges in many elusive reptiles, hence limiting the application of high-throughput sequencing technologies to reptile studies. An alternative reptilian DNA resource suitable for genome sequencing is in urgent need. Here, we used the corn snake (*Pantherophis guttatus*) as a reptile model species to demonstrate that the shed skin is a high-quality DNA source for genome sequencing. Skin sheds provide a noninvasive type of sample that can be easily collected without restraining or harming the animal. Our findings suggest that shed skin from corn snakes yields DNA of sufficient quantity and quality that are comparable to tissue DNA extracts. Genome sequencing data analysis revealed that shed skin DNA is subject to bacteria contamination at variable levels, which is a major issue related to shed skin DNA and may be addressed by a modified DNA extraction method through introduction of a 30 min pre-digestion step. This study provides an enhanced method for the use of reptile shed skins as a high-quality DNA source for whole genome sequencing. Utilizing shed skin DNA enables researchers to overcome the limitations generally associated with obtaining traditional tissue or blood samples and promises to facilitate the application of genome sequencing in reptilian research.

## 1. Introduction

Reptiles exhibit a wide diversity in morphology, behavior, and distribution, making them ideal models for studying the evolutionary processes such as radiation, speciation and adaptation [1,2,3]. Although reptiles represent the most diverse amniote group, they entered the genome era at a pace much slower than mammals. The first complete genome from a non-avian reptile, the green anole lizard, was not available until 2011 [4]. Since then, hundreds of genomes of snakes, turtles, and crocodilians have been completed that have facilitated various genetic studies related to reptilian taxonomy, population genetics, morphological traits, and evolutionary history [1,5,6,7,8,9,10,11,12].

High-throughput genome sequencing routinely requires high-quality template DNA usually extracted from blood or tissue, which, in many cases, is challenging to obtain. The global reptile population is declining with 21.1% reptile species threatened with extinction [13,14]. Researchers are often required by permitting and conservation agencies to use minimally invasive techniques whenever possible, which in some instances may preclude blood or tissue sampling from tail or toe clips. Alternative DNA resources, especially these collected from non-invasive sampling methods, are in urgent need to facilitate future genomic studies of reptiles.

Among the non-invasively collected sample sources available for non-avian reptiles, such as buccal swabs or skin swabs, the shed skin holds the most promise as an alternative DNA resource for genome sequencing. Most reptiles regularly shed off their epidermal outer layers as a coherent sheet [15,16], which can be directly collected from the individuals in captivity or in the field without encountering or handling the animals, since they are usually highly visible. Previous studies have proved that the DNA extracted from shed skins can be applied to PCR-based genetic research [13,17,18,19,20,21]. However, whether reptile skin sheds may yield DNA sufficient for high-throughput sequencing, in terms of quantity and quality, remains to be thoroughly evaluated.

Here, we recruited shed skins from corn snakes (*Pantherophis guttatus*) to assess their utility in high-throughput genome sequencing. With an enhanced DNA extraction method developed from this study that effectively reduced the exogenous DNA contamination from skin sheds, we proved that reptile shed skins provide a suitable source for high-quality DNA that is useful for a variety of genomic applications.

## 2. Materials and Methods

### 2.1. Sample Collection

Eleven shed skins of corn snakes were collected from commercial breeders and hobbyist keepers. Shed skins were preserved in zip-zap bags at room temperature with silicone beads to absorb ambient moisture. Three muscle tissues were collected from three hatches of a corn snake colony kept at Peking University, and frozen immediately at −80 °C. All animal handling and experimental protocols were approved by the Institutional Animal Care and Use Committee of Peking University (IACUC# LSC-LuoSJ-3).

### 2.2. DNA Extraction

Genomic DNA was extracted from 11 shed skin and three muscle tissue samples using the DNeasy^®^ Blood and Tissue Kit (QIAGEN, Valencia, CA, USA). Around 50 mg of shed skin or 25 mg of muscle were used in each extraction. The muscle DNA was extracted following the manufacturer’s instructions. The shed skin DNA was initially extracted following the manufacturer’s manual with a modified incubation time (48 h) and then re-extracted with the introduction of a pre-digestion step prior to the formal extraction steps. The modified DNA extraction method for shed skin was described as follows: (1) A pre-digestion of 50 mg shed skin was first carried out in 360 μL Buffer ATL, 400 μL Buffer AL, and 40 μL Proteinase K at 56 °C for 30 min (or 60 min); (2) the pre-digested shed skin was transferred to a sterile plastic culture dish and washed with 1× phosphate-buffered saline (PBS); (3) the shed skin was further cut into small pieces, and was digested in a lysis buffer containing 360 μL Buffer ATL, 400 μL Buffer AL and 40 μL Proteinase K at 56 °C for 48 h; (4) the digested sample was centrifuged at 14,000 rpm for 3 min, and then the supernatant was harvested; (5) 1/2 volume of ethanol (96–100%) was added to the supernatant, then was mixed by vortex for 30 s, and the mixture was pipetted into DNeasy Mini spin column placed in a 2 mL collection tube; (6) the spin column was centrifuged for 1 min at 12,000 rpm, and the flow-through was discarded; (7) 500 μL Buffer AW1 was added to the spin column, then the spin column was centrifuged at 12,000 rpm for 1 min, and the flow-through was discarded; (8) 500 μL Buffer AW1 was added to the DNeasy Mini spin column, then the spin column was centrifuged at 14,000 rpm for 3 min, and the flow-through was discarded; (9) the spin column was transferred to a new 1.5 mL microcentrifuge tube, and 50–100 μL Buffer AE was added to the center of the spin column membrane to elute the DNA; (10) the spin column was incubated at 25 °C for 1 h and centrifuged for 1 min at 12,000 rpm to harvest DNA. DNA quantity and quality were examined using a Nanodrop spectrophotometer (Thermo Fisher Scientific, Waltham, MA, USA).

### 2.3. Whole-Genome Resequencing and Read Mapping

Whole-genome resequencing was conducted on an Illumina NovaSeq 6000 platform at Novogene Co., Beijing, China. For each DNA extract, a multiplex library with a unique 6-bp index tag was prepared following the standard Illumina library construction protocol (Illumina, San Diego, CA, USA). All libraries with an average insert size of 250–300 bp were sequenced on the Illumina NovaSeq 6000 platform generating 150 bp paired-end reads. The sequencing data for corn snake accessions from this study have been deposited in the NCBI Sequence Read Archive (SRR22317684- SRR22317705; BioProject ID: PRJNA901350).

The adaptor sequences from both ends of the reads and bases with the Phred quality below 30 were trimmed with Cutadapt v1.16 [22]. The Phred quality score (Q score) is the most commonly used metric for assessing sequencing data quality, whereas a Q score of 30 indicates a base call accuracy of 99.99%. The processed reads were aligned to the corn snake reference genome (GCA_001185365.2) with Burrows-Wheeler Aligner v0.7.17 [23]. We used SAMtools v1.7 [24] to filter unmapped reads and sort the mapped reads by coordinates. Then, we used SAMtools v1.7 to estimate the endogenous DNA in each dataset by calculating the ratio of mapped reads and depth to calculated the genome coverage based on the mapped reads of each individual (Appendix A).

### 2.4. Characterization of Contamination Reads

We used SAMtools v1.7 [24] to save all unmapped reads. For samples with unmapped reads at a scale over millions, we used SeqKit v 2.2.0 [25] to randomly extract 100,000 unmapped reads from one sample for further analysis (Appendix A). We used DustMasker v1.0.0 [26] to recognize and filter sequences with low complexity and BLAST+ v2.10.0 (Basic Local Alignment Search Tool) [27] to align unmapped reads with the nucleotide sequence database NT downloaded from NCBI [28]. We filtered out all hits with the E value above 10^−3^. These sequences that match this level of identity are considered strong candidates for contaminants [29]. Only those with over 10 times hits were counted and identified as the sources contaminants, which were classified into Viruses, Eukaryotes, and Bacteria based on NCBI taxonomy [30]. Taxonomic units in Eukaryotes were further categorized into Metazoa, Fungi, and Viridiplantae (Appendix A).

## 3. Results

To evaluate the potential of the reptile shed skin as a DNA resource for high-throughput sequencing, we examined the quality of DNA extracted from 11 corn snake shed skin samples. For comparison, DNA was also extracted from three corn snake muscle tissues as controls. The DNA quality was first evaluated by the Nanodrop spectrophotometer. The 260/280 ratios of the shed skin DNA were 1.83 to 1.93, comparable to that of the DNA extracts from tissue samples (1.84 to 1.99; Appendix A). All the 11 shed skin DNA exhibited relatively high quality with major DNA fragments over 1.5 Kb and only minor to moderate level of degradation (Appendix A) was observed that was comparable to that of tissue DNA extracts. The shed skin DNA concentrations ranged at 79–686 ng/µL (Appendix A), with an average yield of 39.59 µg DNA /100 mg starting sample. This was similar to or even slightly higher than that of tissue samples (35.89 µg/100 mg sample in our test; Appendix A). Overall, the shed skin DNA exhibited high quality comparable to that of tissue DNA, making it an excellent DNA source that could be readily applied to high-throughput sequencing.

We then submitted the 11 shed skin DNA extracts for Illumina library construction and high-throughput genome resequencing, with the three tissue DNA preparations as controls. Each DNA sample was sequenced to an approximate 10× genome coverage. The ratios of bases with the Phred quality score (Q score) ≥ 30 in shed skin DNA sequencing data ranged from 88.38% to 90.78%, only slightly lower than those from tissue DNA data (90.10% to 92.40%) (Appendix A). About 94.00% of the corn snake reference genome (GCA_001185365.2) was covered by the tissue sample sequencing data. Except for PAGU0126 (36.53%) and PAGU0129 (34.62%), similar coverages (92.10% to 93.92%) were also achieved in the sequencing genomes from nine shed skin samples (Appendix A). The tissue samples exhibited consistent endogenous DNA content from 99.17% to 99.48%. The endogenous DNA content from the 11 shed skin samples (Appendix A), however, was variable, with six showing high levels (95.75–98.39%), four medium levels (57.33–71.16%), and one (PAGU0129) a low level of endogenous DNA content (5.84%). Overall, these results suggest that shed skin can serve as a DNA resource for high-throughput sequencing, though it is crucial to develop a procedure to effectively reduce exogenous DNA contamination.

To this end, we performed BLAST analyses on the unmapped reads from shed skin sequencing data to elucidate the sources of the exogenous contamination. About 58.64% of the unmapped reads were assigned to Viruses, Bacteria, or Eukaryota. Specifically, 42.01–70.20% of the unmapped reads from shed skin sequencing data were assigned to Bacteria, while such a proportion in the tissue sequencing data was 12.02% in average (Appendix A). This indicates that the major exogenous contamination of corn snake shed skin is from bacteria. Additional sources of contamination in the snake skin sheds included viruses and eukaryotes (such as fungus), with 0–3.72% and 0.29–7.12% of the unmapped reads aligned with Viruses and Eukaryota, respectively (Appendix A, Figure 1).

Given that the corn snake shed skin was predominantly contaminated with bacteria, which was most likely attached to the surface, we introduced an additional pre-digestion procedure before proceeding to the DNA extraction steps, to reduce the exogenous DNA contamination. Based on sequencing results, four shed skin samples representing different levels of exogenous DNA contamination were selected to test the modified DNA extraction method (Appendix A, Figure 2). The sample set included one with a high level of contamination (PAGU0129), two with a medium level (PAGU0093, PAGU0126), and one with a low level of contamination (PAGU0092). Each sample underwent two independent DNA extractions with the pre-digestion step of 30 and 60 min, respectively.

All eight shed skin samples extracted using the modified method yielded higher-quality DNA with minor to moderate degradation. Major fragments were over 3 Kb (Appendix A), which was nearly double the fragment length (1.5 Kb) using the original extraction method and hence well-qualified for high-throughput sequencing. We submitted the eight DNA samples for sequencing, generating approximately 2 Gb of data to assess the effectiveness of the modified extraction method in reducing exogenous contamination.

Overall, the sequence accuracy of these eight DNA samples was comparable to that of tissue, with 88.72–91.20% of the base pairs showing Q scores greater than 30. For the sample PAGU0092 showing low level of contamination from the first run, the endogenous DNA content remained consistently high (99.06%, 99.53%) using the modified DNA extraction method (Appendix A, Figure 2). For PAGU0093 and PAGU0126 with medium levels of contamination, the introduction of a 30 min pre-digestion step resulted in an increase in endogenous DNA content (71.16–78.06%; 59.86–98.91%), which remained stable when the pre-digestion was extended to 60 min (Appendix A, Figure 2). For the highly contaminated sample PAGU0129, the endogenous DNA content was substantially enhanced, rising from 5.84% to 33.68%, after the introduction of a 30 min pre-digestion step, but fell back to 9.86% if the pre-digestion step was further extended to 60 min (Appendix A, Figure 2). In conclusion, a 30 min pre-digestion appears to be the optimal step to effectively eliminate the exogenous DNA contamination in the corn snake skin shed samples.

## 4. Discussion

Most non-avian reptiles periodically shed off skins as part of their natural development process. These skin sheds left behind in the environments or cast away by their breeders in captivity are readily available for sampling, representing perhaps the most accessible biological material from reptiles. In this study, we proved the reptile shed skin as a promising non-invasive DNA resource and modified the extraction method for an enhanced genomic DNA quality that was comparable to tissue DNA and could be directly applied for high-throughput sequencing. This shed-skin-based approach promises to facilitate reptile studies in the genomic era, especially for those rare, elusive, or endangered reptilian species, whose tissue or blood samples are difficult to obtain [18].

We noticed that shed skins of corn snakes tested in the study were predominantly contaminated by bacteria, while levels of contamination varied across samples. This variation may be attributed to the condition of the shed skin prior to collection. Newly shed skins of corn snakes are usually moist, along with the warm temperature required by reptiles’ natural history, forming a suitable environment for bacteria to grow. The longer the shed skins are exposed to humidity and temperature, the more severe they are subject to bacteria contamination. Hence, it would be ideal to collect and desiccate the shed skins as soon as they shed off, to minimize degradation.

We introduced a pre-digestion step of 30 or 60 min in length and evaluated their effect in eliminating the bacteria contamination in corn snake skin sheds. For the samples with minor to medium levels of contamination, either 30 or 60 min pre-digestion is efficient in reducing exogenous DNA content. In the case of a severely contaminated sample (PAGU0129), the 30 min pre-digestion performed well and substantially increased the endogenous DNA content, whereas a 60 min pre-digestion did not show a significant effect of improvement. It is likely that the sample had been left in a warm and moist condition for a long time before collection, which resulted in not only severe bacteria contamination, but also a considerable level of degradation. As degraded samples are subject to digestion, a prolonged (60 min) pre-digestion procedure aiming at removing contamination may likely simultaneously lyse a large portion of degraded shed skin, whose endogenous DNA was washed away during the subsequent wash step. Overall, a 30 min pre-digestion is recommended as the optimal duration for processing shed skin samples, which would effectively eliminate bacteria contamination without compromising endogenous DNA.

## Figures and Tables

**Figure 1 genes-14-01678-f001:**
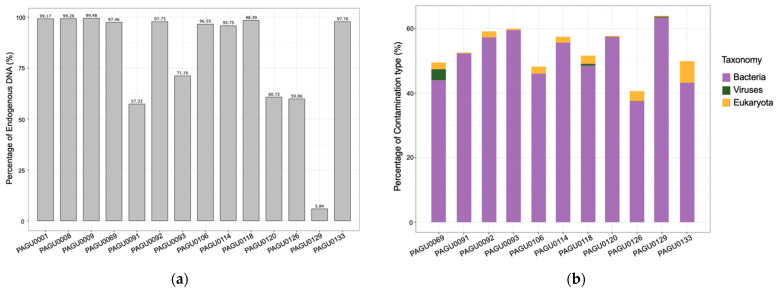
The DNA composition of extracts from tissue and shed skin samples of corn snakes. (**a**) The content of endogenous DNA in the extracts from corn snake tissue and shed skin samples. PAGU0001, PAGU0008, and PAGU0009 were from tissues, and the others from shed skins. (**b**) The composition of exogenous DNA (contaminations) in the shed skins DNA extracts. DNA reads with uncertain taxonomy origin were not shown.

**Figure 2 genes-14-01678-f002:**
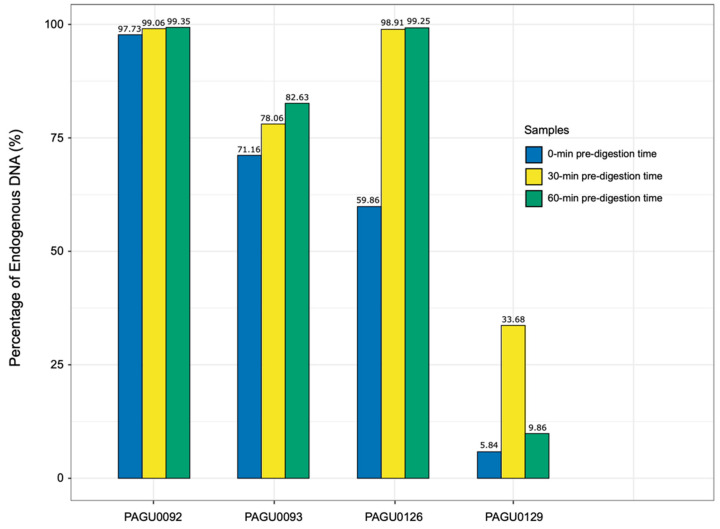
The comparison of endogenous DNA content of shed skin DNA using different extraction strategies.

## Data Availability

The sequencing data for corn snakes accessions from this study have been deposited in the NCBI Sequence Read Archive (BioProject ID: PRJNA901350).

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
