# Peer review of "An Enhanced Method for the Use of Reptile Skin Sheds as a High-Quality DNA Source for Genome Sequencing"

_genes, 2023, doi:10.3390/genes14091678_

Round 1
Reviewer 1 Report
Studies like this are extraordinarily useful to the scientific community. The authors present a simple modification to kit-based genomic DNA isolation involving an additional digestion step that appears to improve the quality of extractions from snake sheds.
I have some relatively minor suggestions for the manuscript. I have tried to identify areas where the text of the article may be grammatically improved. My edits are suggestions to improve the flow of the article. The authors should feel free to adopt those changes as they see fit or modify them accordingly.
Page 1, 2nd paragraph – The sentences in this paragraph could be broken up for clarity and have some grammatical mistakes. A potential suggestion for this paragraph is as follows.
High-throughput genome sequencing routinely requires high-quality template DNA usually extracted from blood or tissue, which, in many cases, is challenging to obtain. The global reptile population is declining with 21.1% reptile species threatened with extinction [13, 14]. Researchers are often required by permitting and conservation agencies to use minimally invasive techniques whenever possible which in some instances may preclude blood or tissue sampling from tail or toe clips. Alternative DNA resources, especially these collected from non-invasive sampling methods, are in urgent need to facilitate future genomic studies of reptiles.
Page 2, 1st paragraph – The authors cited Brekke et al.2019, Xu et al. 2020, James 1999, and Rajpoot et al. 2021 referencing prior studies documenting DNA isolation techniques from sheds. However, there is also Eguchi and Eguchi 2000 Biotechnology Letters 22:1097-1100 and Bhaskar et al. 2022 Journal of Wildlife and Biodiversity 7: In Press. Additionally, the author of citation 18 is actually James W. Fetzner Jr of the Carnegie Museum of Natural History in the United States and should be cited as Fetzner J. W. and not James W. F. The authors should double check their citations for any additional mistakes.
Page 2, 3rd paragraph – The end of the second sentence should read “…moisture”, not “…moist”.
Page 2, 4th paragraph – Are the buffers ATL and AL the names of the buffers included in the Qiagen DNeasy kit? If not then the recipes for making these buffers should be included in the paper, perhaps in an appendix. Was there any initial shredding or mincing of the shed prior to the digestion step (step 1)? I will note that Eguchi and Eguchi 2000 added collagenase to a similar process and obtained a higher yield of genomic DNA.
Page 3, 3rd paragraph – At the end of the fourth sentence the -3 in “10-3” should be a superscript.
Page 3, 4th paragraph – Quantification was done with a Nanodrop spectrophotometer. The Nanodrop instrument is notoriously inaccurate compared to fluorescence-based methods of quantification such as the Qubit instrument. However, the Nanodrop is very useful in determining DNA quality through the absorption ratio of 260 nm / 280 nm with a ratio of approximately 1.8 as being indicative of “pure” DNA (see ThermoFisher Scientific’s technical bulletin on the Nanodrop instrument). If Qubit measurements or Nanodrop 260/280 ratios were available this might be very useful in comparing the quality of the genomic DNA from sheds versus those isolated from tissue.
Page 4, 2nd paragraph – Was there also an attempt to BLAST the unmapped reads from the tissue samples? This could be an interesting comparison.
Supplementary figures – I will note that the quality of the supplementary figures is somewhat poor and the text on the axes is difficult to read.
I think the overall quality of the English is quite good but there are a few spots where there could be improvements. I offered some suggestions for the grammar in specific sections in my comments to the authors.
Reviewer 2 Report
Dear all
The article is extremely important, as there are few researchers working with wild animals. Recommend its publication with minor revisions.
In the article, they use the changed skin of the snake to extract DNA, comparing it with muscle tissue samples. Did these muscle tissue samples follow any animal care protocol? Was the animal sacrificed later or not, what was the fate of the animal?
I understand it was invasive, so it would need some authorization, protocol, etc.
Therefore:
- Add Animal Ethics Statements in Methods:
1) name the institutional animal care review committee that approved the study;
2) name of the international animal care guidelines that were followed;
3) state the fate of animals used in the experiments. For animal sacrifice, include the method and compounds used.
Also the BioProject ID: PRJNA901350, was not found in NCBI.
